# Differences across Playing Levels for Match-Play Physical Demands in Women’s Professional and Collegiate Soccer: A Narrative Review

**DOI:** 10.3390/sports10100141

**Published:** 2022-09-22

**Authors:** Erin Choice, James Tufano, Kristen Jagger, Kayla Hooker, Kristen C. Cochrane-Snyman

**Affiliations:** 1School of Physical Therapy, Regis University, Denver, CO 80221, USA; 2Faculty of Physical Education and Sport, Charles University Prague, 16252 Prague, Czech Republic; 3Department of Health and Human Performance, Concordia University of Chicago, River Forest, IL 60305, USA; 4Department of Nutrition and Exercise Physiology, Washington State University, Spokane, WA 99202, USA

**Keywords:** female, GPS, external load, workload, athlete management, training program design, collegiate sport, soccer, match play, football

## Abstract

Advancements in sport technology have made quantifying match-play external load (e.g., total distance, sprint distance, number of sprints) a popular option for athletics personnel. These variables of volume and intensity are useful for both objectively monitoring training in field-based sports and for designing training programs. As physical abilities differ across playing levels, match-play demands likely also differ. This narrative review compiles and compares the match-play external load data for women’s soccer at the professional and collegiate levels. Databases were searched through July 2022, yielding 13 primary articles that assessed the match-play demands of women’s soccer (3 professional, 8 Division I, 1 Division II, and 1 Division III). The results indicate that the average total distance covered were similar between the professional, Division I and Division III levels, but the variability was greater among Division III compared to professional and Division I players. Data for Division II are scarce, but the total distance covered appears to be less than for professional, Division I and Division III. There was also large variability for sprint distance and number of sprints across data at all playing levels. Considering the lack of studies of Division II and Division III players, more research is necessary to determine how playing level may affect external load profiles, as isolated studies likely only reflect data from isolated teams.

## 1. Introduction

In order to implement evidence-based training practices and optimize training stress, understanding the physical demands of a sport is critical. Considering the sport of soccer, match-play data can provide objective measures of match-play demands, can reveal positional differences during match play, can help coaches identify changes in performance over time, and can help guide player management strategies [1,2,3,4,5,6,7,8,9,10,11,12,13]. Objective data for match-play demands (also referred to as external load, which includes total distance, sprint distance, top speed, number of sprints/explosive efforts, and Player Load, among others) are relatively easy to capture with GPS microtechnology which has become commonplace in professional and collegiate settings.

The physical demands of professional and collegiate male soccer athletes have been studied extensively for over 35 years [1,10,14,15,16,17,18,19,20,21,22,23]. Less information exists regarding the physical demands of female soccer athletes at the professional level [11,24,25] and NCAA Division I (DI) level [10,12,26,27,28,29,30], and even less evidence exists specific to NCAA Division II (DII) [2] and NCAA Division III (DIII) [31,32]. The lack of research on female players at each playing level substantiates the need for a narrative review that explores and summarizes match-play demands across playing levels for women’s soccer.

Recent research on female soccer athletes, including differences across playing levels, has developed due to findings of differences in aerobic fitness between playing levels in men’s soccer athletes from youth to the elite [19,22,23]. More advanced male players on competitive U14 and U16 teams have greater VO_2_max values than less advanced players, and aerobic fitness was a discriminator of successful performance between playing levels [22]. Among different levels of elite players (professional players on an elite European team, professional players on the highest German league, and players on the German fourth league), performance on intermittent recovery endurance tests was better at the higher levels of play [19,23].

Fitness differences across playing levels are evident and expected, but understanding fitness levels alone is not adequate for programming for the match-play demands of the season. Training programs should be designed with the end goal in mind, which is why match-play data are important for the design of the annual periodized training program. Average distance covered, sprint distance, top speed, and number of sprints, as well as any positional differences and differences across time, are all valuable data when designing a training program. It is understood that soccer involves both the aerobic and anaerobic energy system pathways. Average distance covered will impact volume of running (aerobic training), while sprint distance, top speed, and number of sprints will impact volume and intensity of sprint training (anaerobic training). These data will be specific to teams and level of play and will allow for optimal supercompensation rather than overtraining or undertraining for match-play throughout the season.

Research is now trending to explore match-play data at various playing levels for this reason. Recent research supports the argument that there are differences in match-play demands across playing levels in female soccer [9,33,34]. A 2008 study [33] determined top-class international players complete more high-intensity running intervals compared to lower standards of play during match-play. A 2021 perspective review [9] focused on the importance of training appropriate attributes in order to prepare female soccer athletes to play at progressively higher standards due to a linear increase in total distance covered from youth to NCAA to professional level match-play. Finally, a 2022 female soccer systematic review [34] revealed that physical characteristics of match-play (e.g., total distance) do appear to increase between playing standards (from youth U13 to professional) and differences exist between field positions.

The purpose of this review is to consolidate available match-play data for each playing level to reveal similarities, discrepancies, and gaps to shed light on the need to create external load profiles for female soccer athletes at specific playing levels. External load profiles are imperative for coaches to program appropriately for the expected demands of match-play. A secondary purpose is to suggest future directions for research, which could help provide coaches with the data necessary to better design and implement evidence-based training programs.

## 2. Materials and Methods

### 2.1. Search Strategy

The databases Google Scholar and PubMed were searched using a combination of the following search terms to capture all relevant studies: “external demands”, “physical demands”, “women’s soccer”, “female soccer”, “college”, “division II”, “soccer”, “female soccer during matches”, and “performance of women’s soccer players”. The reference lists of related papers for other relevant articles were also used [4,5,15]. The search strategy was independently performed by the lead author for articles published between January 2000 and July 2022.

### 2.2. Study Selection

Inclusion criteria were based on the PICO framework: population, intervention, comparison, and outcomes. Only observational studies including match-play data (via GPS 5 Hz, 10 Hz or 20 Hz; triaxial accelerometers; or time-motion analysis via video recording) for professional or collegiate (DI, DII, DIII) female athletes were used. Studies must have analyzed match-play data for more than one regular season match. Any single variable or any combination of the following objective match-play variables were included: total distance, sprint distance, speed, Player Load, and number of sprints. Studies must have included objective match-play data for live soccer competitions. Data from the 2020 season, which was modified due to COVID-19, were excluded. Non-soccer sports data were used for comparison, but were not the primary articles selected for the review. After the initial search was completed, 65 articles were identified for further review. These articles were examined for alignment with the PICO criteria and assessed for quality and inclusion criteria by the primary and secondary authors. Thirteen total original articles examining soccer athletes were identified with direct alignment with the PICO standards for inclusion within the narrative review.

### 2.3. Data Extraction

The following information was extracted from the selected studies: author and year of publication, sample (number of participants and level of play), player positions (only field positions were included, goalkeepers were excluded), length of time data were collected, equipment used to capture match-play data, and reported average (Mean ± SD) external load values (total distance, sprint distance, number of sprints, maximum speed, Player Load). Positional differences (if any) were noted as well as any results that described changes over time (between halves or between days of the week). The main conclusion statements of each publication were also summarized to organize the main significant results of each study. Data were extracted into standardized spreadsheets and are summarized in Table 1.

## 3. Results

### 3.1. Demands of Match-Play at Each Level

TOTAL DISTANCE: Total distance during match-play for professional athletes ranged from 9040 m ± 938 m [11] to 10,500 m ± 800 m [24] with positions combined. Total distance for DI athletes was similar to or less than that of professionals, and had similar or greater variation (values were as follows: 9486 m ± 300 m [28], 9058 m ± 840 m [26], 7482 m ± 959 m [30], 8310 m ± 1400 m [10], and 8320 m ± 1963 m [27]). Total distance during match-play for a DII team was 5480 m ± 2350 m [2], whereas 9807 m ± 2588 m for a DIII team was reported [31]. Results are presented in Figure 1.

SPRINT DISTANCE: Mean sprint (≥18 kph) distances for professional athletes have been reported to be 657 m ± 157 m for forwards, 447 m ± 185 m for midfielders, and 545 m ± 217 m for defenders [25], while sprint (>15 kph) distance has also been reported as 1108 m ± 294 m with positions combined [11]. High-intensity running (HIR) (>15 kph) has been reported to be 1310 m on average [24]. Total sprint distances covered during match-play for DI athletes varies throughout the literature, ranging from 86 m ± 81 m [30] to 401 m ± 158 m [10] to 747 m ± 387 m [27] per match, due to differing definitions of speeds designated as a “sprint” (definitions include ≥22 kph [30], >10 kph [10], >18 kph [27],). Per position, forwards had sprint (>18 kph) distances between 608 m ± 199 m [26] and high-speed running (>15 kph) distances of 682 m ± 492 m [12], midfielders covered between 256 m ± 75 m [26] and 595 m ± 428 m [12], and defenders had the widest range with 277 m ± 61 m [26] to 734 ± 367 m [12] per match. The larger variation in sprint distances for midfielders and defenders is due to different definitions of sprint/high speed and player descriptions. One study [12] aggregated player positions into only three groups—forward, midfield, and defense—while the other [26] more specifically distinguished between positions—forward, wide midfield, central attacking midfield, central defending midfield, fullback, and central defense.

There are no available match-play data for mean sprint distance for DII athletes. Division III athletes sprinted 1019 ± 552 m per match, defined as “high speeds” that were ≥15 kph, with outside midfielders covering the most distance at high speeds [31]. Results are presented in Figure 2.

NUMBER OF SPRINTS/EXPLOSIVE EFFORTS: At the professional level, forwards have been reported to sprint 43 ± 10 times per game, midfielders 31 ± 11, and defenders 36 ± 12 [25], with an average of 26 sprints for all field positions combined [24]. Number of accelerations for professional players have also been reported to be 225 ± 40 [11]. For DI players, data indicate that they perform 14 ± 5 sprints per match (>10 kph) [10], while other studies have determined that they perform 33 ± 17 [27] and 31 ± 13 [28] sprints (>20 kph) per match, or even fewer at 4 ± 4 (≥ 13 kph) [35] per match. There are no available match-play data for the number of sprints or explosive efforts for DII athletes. Number of sprints have been recorded to be 15 ± 8 (>19 kph) for DIII athletes. Results are presented in Figure 3.

### 3.2. Positional Differences (Regardless of Playing Level)

Midfielders have been shown to cover the greatest distances during match-play compared to forwards and defenders in multiple studies [24,25,26,31]. However, forwards have been shown to cover significantly more distance than midfielders and defenders in a different study [28], and forwards have been shown to cover greatest distances at all speeds compared to midfielders and defenders [28]. Additionally, defenders covered less total absolute distance than midfielders in the first half, and less total distance than midfielders and forwards in the second half [29]. Contrarily, another study showed that defenders covered greater total distance than both forwards and midfielders [12]. In contrast to these collective aforementioned findings, a different study determined there were no positional differences (forward, midfielder, defender) in total distance covered during match-play [27].

### 3.3. The Influence of Time (Friday to Sunday Match) on Objective External Load

Current evidence indicates that NCAA DI female soccer athletes cover less distance of HIR in matches played less than 48 h after another match [35]. Data indicate that no differences were found in minutes played, distance rate, or number of sprints between Friday and Sunday matches, but a significant decrease in rate of HIR (defined as running at a velocity equal to or exceeding 4 m·s^−1^ for longer than 1 s) between Friday (25 ± 7 m·min^−1^) and Sunday matches (23 ± 6 m·min^−1^) was observed in a DI women’s soccer team [35].

### 3.4. The Influence of Time (First Half to Second Half) on Objective External Load

Current evidence reveals that female professional level soccer forwards had larger reductions in the number of sprints and total sprint distance from the first half to the second half of the match compared to midfielders and defenders [25]. Performance comparisons between the first and second half within matches have also revealed a significant decrease in high-speed and high-intensity runs during the second half of the postseason compared with the regular season in NCAA DI women’s soccer [30].

Seventy-one professional players competing in full matches during twelve regular season matches were assessed for differences in sprint distance by half, and there was a 21% reduction for forwards and a 5–9% reduction for midfielders and defenders [25]. This outcome is also reflected in the reduced number of sprints for forwards (24 ± 7 vs. 20 ± 6 sprints) [25].

## 4. Discussion

The exploration and comparison of these 13 articles was guided by the known differences in physical abilities across playing levels in both male and female soccer athletes [9,19,22,23,32,33,34], and the known rule differences between professional level and collegiate level play [36,37]. The evaluation of match-play external load data across playing levels in female soccer is necessary to eliminate some of the guess work for coaches and provide more objective data specific to level of play for female soccer athletes in order to train these athletes adequately for competitive season match-play. However, the lack of available data made this comparison challenging, as many variables of match-play have not been studied extensively at each level of play.

### 4.1. Total Distance

Based on the most comparable available data for DI [28], DII [2], and DIII [31], mean total distance for DI and DIII is similar (9486 m and 9807 m per match, respectively), while DII mean total distance is meaningfully less (5480 m per match). The data from the DI [28] and DIII studies [31] represent average starting players, while the DII data include all players—both starting players and reserves [2]. Additionally, there is variability within the inclusion criteria for minutes played: all minutes [28], 50% or more of match minutes [31], and any number of minutes [2]. Future DII research should match similar higher minute inclusion criteria as past DI and DIII studies in order to capture the match-play demands for the average starting player [10,26,27,28,30,31], as there are many DII rostered athletes who do not play a significant number of minutes (reserve players) [38]. Inclusion of data from the reserve players who play fewer minutes skews the team averages, which is an important consideration for both DII and DIII team data due to the differences in skill level across athletes on one team. If data for all players need to be captured and analyzed, they should be normalized per minute because of the impact of number of minutes played on overall external load data. Additionally, the available DII match-play data only include average total distance [2], which is not sufficient when comparing the available data that represent DI and professional athletes. This leaves coaches who work with DII women’s soccer athletes (7600 athletes nationwide [39]) at a disadvantage, with less match-play external load data available, for making evidence-based decisions for training programs.

Mean total distances are similar for DI and DIII athletes when comparing the Sausaman et al. (2019) [28] and Jagim et al. (2020) [31] studies, but it is important to note that DI athletes had less variability in total distance covered (± 300 m) [28], whereas the findings amongst the DII and DIII athletes had more variability among the athletes (± 2350 m and ± 2588 m, respectively) [2,31]. Among other studies by McFadden et al. (2020) [10] and Corrales (2020) [27], the variation for DI athletes remained less than that found in the DII [2] and DIII [31] studies (± 1963 m [27] and ± 1400 m [10]) but the values were slightly more comparable. It is also important to consider the threshold for the number of minutes played that was used by different studies. A study that includes data only from starters who played the entire 90 min [28] would be expected to have a different variation in the total distance than studies with less restrictive thresholds that included all athletes who played 79 ± 11 min [27] or 79 ± 4 min [10].

### 4.2. Positional Differences in Total Distance

Most of the data pertaining to positional differences is specific to professional and DI playing levels, as these data are minimal for DIII and absent for DII players. Positional difference findings (Table 1) demonstrate longer total distance demands for some positions on some teams—without consistency of position across studies. Research available to date, however, does not take into consideration formations on the field (i.e., 4-3-3 vs. 3-5-2), which could greatly impact expectations for total distance covered by each position.

### 4.3. Sprint Distance

Specific attention should be given to sprint distance and high intensity running (HIR) in the sport of soccer due to the anaerobic component of the sport. Sprinting performance and match-play HIR appear to be important predictors of physical performance in female soccer athletes and HIR has been shown to be positively correlated with ${\;\dot{V}O}_{2Max}$ [24,25,30,33,40,41,42,43], yet little is known about how much time and distance, on average, are spent sprinting or running at high-intensity during match-play. Substantial variability exists within the data due to differing definitions for “sprint”; with definitions varying from speeds greater than 10 kph to speeds greater than 22 kph. The significant variability in the available literature does not allow for generalized conclusions to be drawn. The variability in sprint distance covered could be due to the number of shots on goal, physical characteristics of the athletes, substitution strategies, or playing style, and could be used by coaches to inform training programs.

### 4.4. Changes across Time (Friday to Sunday and First Half to Second Half)

Current evidence indicates that NCAA DI female soccer athletes cover less distance of HIR in matches played less than 48 h after another match [35]. This is relevant and an important consideration because collegiate women’s soccer regular season conference play most commonly includes two matches per week that are roughly 42 h apart: Friday evening and the following Sunday at mid-day. The efficacy of a 42-h recovery period between the first match of the week and the second match of the week, and the changes in performance that occur from first half of match to second half of match are not well understood at any playing level. The current evidence supports NCAA DI and professional players only, with only three studies on this topic dedicated to female soccer athletes [25,30,35].

Player Load, number of sprints, and sprint distance have all been shown to decrease from the first half to the second half of match-play [1,25,30,34,44], and further research is warranted to determine if this is due to fatigue levels, in which case fatigue could be managed differently to alter the outcome, or if this is due to number of goals scored, competition level, or tactical strategies used (all variables that have not been assessed in the available literature). With further research to increase understanding of why a decrease in external load variables is seen from first half to second half of the match, training programs and coaching strategies can be altered accordingly.

Similar to women’s soccer studies published in 2012–2016 [25,30], a study conducted with NCAA DI women’s lacrosse athletes found significant decreases in sprint distance and number of Power Plays (a type of explosive effort derived from PlayerTek technology) from first half of the match to second half of the match [44]. The data from the DI lacrosse study demonstrated that only midfielders had a significant decline in sprint distance with both midfielders and defenders significantly decreasing the number of Power Plays from the first half to the second half of match-play, indicating a connection between physiological fatigue and the ability to express peak fitness capabilities [44]. A similar study should be conducted with women’s soccer to determine what trends are evident from first to second half to further inform decision making and program design by women’s soccer coaches.

It is also important to consider how much total distance players experience during each match, especially considering total distance is a strong predictor for session rating of perceived exertion (sRPE) [12]. Among matches against ranked opponents during an NCAA DI female soccer season, those with the fewest substitutions (three or fewer throughout the match) were found to also be the matches that were lost, while the match that was won involved five substitutions [13]. When there are more substitutions, each player’s external load is expected to decrease, thus allowing a greater opportunity to recover and perform more optimally. The management of playing time and training programs (e.g., conditioning programs to improve ${\dot{V}O}_{2Max}$) for NCAA soccer athletes may be a key factor affecting running performance during competition; and increased understanding of these changes may assist with the overall management of player fatigue and performance, given the typical match schedule parameters for these athletes.

### 4.5. Summary and Limitations

While there were differences noted in match-play metrics between NCAA DI, DII, and DIII female soccer athletes, it is difficult to draw clear conclusions on physical demands of the varying levels of play due to an insufficiency of available GPS data at all levels. Without a collection of data from each level of play, the few studies that do exist may misrepresent the typical physical demands experienced, and therefore mislead the generalizable conclusions. For example, prior to the addition of the most recent professional level article in this review [11], it would have appeared that professional female soccer athletes run a greater total distance than their NCAA DI counterparts; however, with the addition of the newer article, the values for the two levels are much more comparable. This illustrates the importance of the ongoing effort to establish match-play demands at varying levels of play, especially as sport technology continues to advance and becomes more utilized in athlete management strategies.

It is important to acknowledge two major limitations in the current literature. Very few studies present the data relative to time played in a match. The data are based on total volume for a match regardless of time played. Second, there are references to different metric definitions and thresholds within the literature. Further delineation regarding the use of absolute versus relative thresholds in the literature and the use of uniform speed zones is a significant limitation when making comparisons.

### 4.6. Practical Application

Adequate training includes training specifically for match-play demands and incorporating proper overload and taper [3]. Coaches must implement proper overload and taper, and these external load variables must be measurable above and below match-play demands, respectively. With evidence to show that some DIII athletes cover greater maximum distances than DI athletes [28,31], coaches can improve efficacy of training, especially in the off-season and pre-season, by implementing training programs that account for the known expected maximum and mean match-play distances. Additionally, training programs can be considered more effective if tailored to the demands of the starting players, rather than including data for the reserves. This should include appropriate overload principles in the off-season for the less fit athletes in order to avoid non-functional overreaching or overtraining. Further, match-play data may aid coaches in making informed roster and in-match decisions. For example, considering the associations noted between substitutions and match result [13], it may be worth making more frequent substitutions throughout a match; similarly, aiming for a slightly larger roster may enable more substitutions to be made without a large change in expected performance as players rotate through.

Acquisition of team specific match-play data can also lead to improved decision-making by strength and conditioning coaches. For example, if team data reveal that midfielders run significantly longer distances than forwards and defenders, as found from team-specific data for females [25,26,31] and for males [1], midfielders may need altered or more aggressive recovery strategies due to higher volumes (longer distances covered) during match-play, as well as higher training volumes during pre-season training in order to adequately prepare for the competitive season’s expected volume. Similar training decisions can be made if total distance, sprint distance or number of sprints in the second half is unexpectedly low compared to the first half, or lower during the Sunday game compared to the Friday game as found in available data [1,25,30,44]. Additionally, if team-specific data reveal forwards run significantly faster top speeds compared to midfielders and defenders, coaches may consider that forwards should be completing more speed-specific training during off-season and pre-season in order to fulfill their role optimally during the competitive season.

## 5. Conclusions

Bridging the gaps between performance analysis research and practical application in sport requires analysis of performance at various levels of play, and is imperative for advancement in the field. With the relatively easy method of data collection through GPS microtechnology, and the current known gaps in the literature regarding match-play demands across playing levels for women’s soccer, data collection efforts should increase to allow for more comparison and application. Match-play activity profiles based on biological sex, position, level of play and changes in these profiles across time supply information to coaches about what might be expected of teams as a group, and specific athletes as individual performers during match-play. Analysis of match-play activity profiles can also reveal opportunities to train or recover more effectively for optimal match-play throughout the duration of the competitive season. Ultimately, the research pertaining to elite female soccer athletes has substantially increased over the past decade, but there are still gaps in what athletics personnel know about athletes specific to level of play (professional, DI, DII, DIII). There is currently published evidence to describe the average starting players’ match-play demands at the professional, DI, and DIII levels, but DII match-play data are lacking, and more research is warranted to fill this gap, as there are significant differences in reported physical abilities across levels of play for female soccer athletes [9,32,33,34]. The current published data have attempted to create an external load profile for the female soccer athlete specific to playing level (professional, DI, DII, DIII) and most available data are team-specific data, rather than generalizable data for each playing level. The addition of more match-play data to the body of literature, and team-specific analysis of match-play data derived from GPS microtechnology, will allow strength and conditioning coaches and soccer coaches to collaborate to make better evidence-based decisions for training and recovery in order to optimize match-play.

Ultimately, coaches should consider match-play external load averages and differences among playing positions (if any), as well as changes in performance as a function of time (first to second half of match and Friday to Sunday match). Collectively, these considerations can inform decisions about how to mitigate undesired changes in performance, and how to assess whether recovery is sufficient. Additionally, these considerations can inform how the athletes are individually responding to the external demands. The combination of these variables and the results presented should be used to take a holistic approach to athlete management in order to optimize training programs and performance for competitive female soccer athletes at specific levels of play.

## Figures and Tables

**Figure 1 sports-10-00141-f001:**
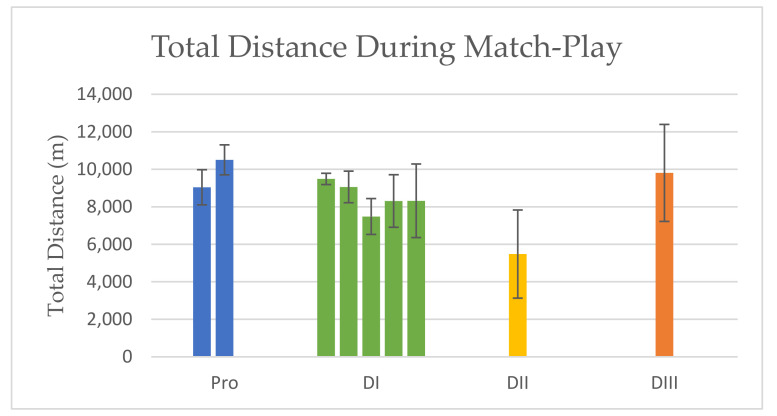
Total Distance During Match-Play. Data are mean ± SD.

**Figure 2 sports-10-00141-f002:**
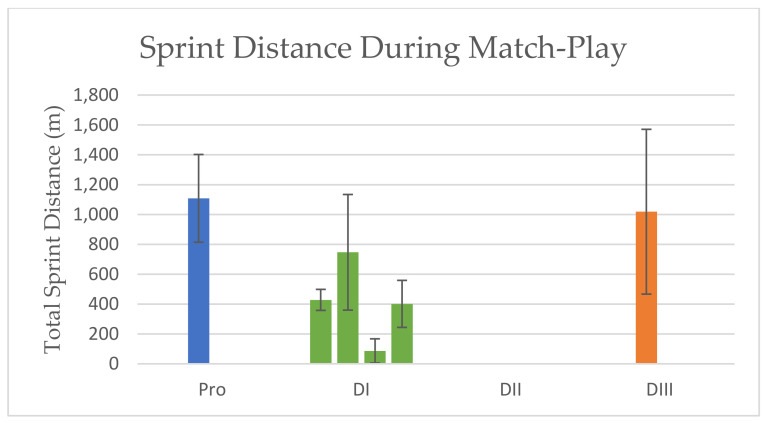
Sprint Distance During Match-Play. Data are mean ± SD. Left to Right Sprint Definitions are as follows: >15 kph, >18 kph, >18 kph, >22 kph, >10 kph, ≥15 kph.

**Figure 3 sports-10-00141-f003:**
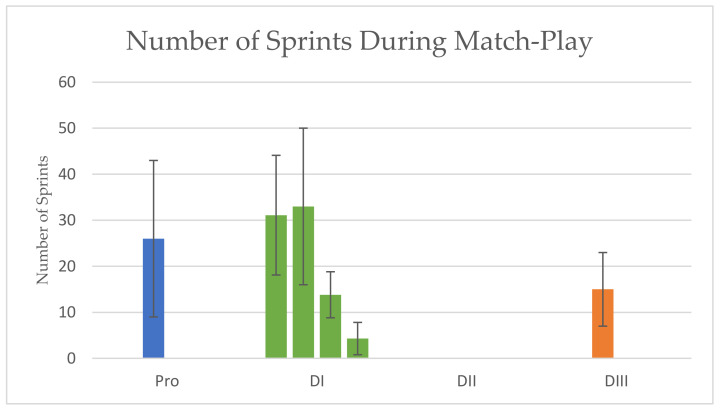
Number of Sprints During Match-Play. Data are mean ± SD.

**Table 1 sports-10-00141-t001:** Match-Play Physical Demands of Women’s Soccer.

Author	Sample	Design	Equipment	Results: Total Distance (m)	Results: Mean Sprint Distance (m)	Results: Number of Sprints	Results: Across Time	Conclusion Statement
Vescovi [25]	71 ProF: 16 MF: 26 D: 29	RA across 12 regular season	5 Hz GPS—SPI Pro, GPSports	DNE	F: 657 ± 157 MF: 447 ± 185 * D: 545 ± 217	F: 43 ± 10 * MF: 31 ± 11 D: 36 ± 12	DNE	These data should guide the development of high intensity and sprint thresholds for elite female soccer players
Krustrup et al. [24]	14 Pro	RA across 4 regular season	Time-motion analysis: VHS movie camera (NV-M50, Panasonic)	10,500 ± 800	AVG HIR: 1310	26 ± 17	Distance covered at a high intensity decreased from first to last 15-min period of the first and second halves **	Match performance of elite female soccer players varies with differences in the physical capacities of the players
Romero et al. [11]	18 Pro F: 2 MF: 10 D: 6	RA across 1 competitive season	5 Hz GPS—SPI Pro X, GPSports	9040 ± 938	1108 ± 294	DNE 255 ± 40 (# of accelerations)	DNE	Match load was greater than training load and differed by position while training load did not. Differences between positions should be accounted for when prescribing training load to optimize performance and reduce injury risk.
Sausaman et al. [28]	23 NCAA DI	RA across 4 competitive regular seasons	10 Hz GPS—Catapult Systems	Combined: 9486 ± 300 -------------- F: 9882 * MF: 9536 D: 9039	Combined: 428 ± 70 ------------- F: 633 * MF: 267 D: 385	Combined: 31.1 ± 13	DNE	The physical demands of DI soccer differ * by position and appear lower compared to higher standards of play
Alexander [26]	10 NCAA DI	RA across 1 competitive season	10 Hz GPS—Catapult Systems	Combined: 9058 ± 840 ---------------- F: 9695 ±401 * Wide MF: 9500 ±847 CA MF: 9236 ± 491 Central D MF: 9947 ± 578 * Fullback: 9306 ± 377 Central D: 8041 ± 371 *	F: 608 ± 199 Wide MF: 518 ± 154 CA MF: 316 ± 105 Central D MF: 256 ± 75 Fullback: 467 ± 147 Central D: 277 ± 61	F: 47 ± 4 * Wide MF: 36 ± 9 * CA MF: 23 ± 6 Central D MF: 18 ± 5 Fullback: 39 ± 6 * Central D: 19 ± 3	High-speed efforts dropped approx. 18% from the first match to the second match on average **, and distance covered in high speed dropped by approx. 8% **	Significant differences exist between positions for total distance covered during a match *. Forward and central defensive midfielder positions covered the greatest distance during match play
Corrales [27]	10 NCAA DI F: 3 MF: 3 D: 4	RA across 8 regular season matches	10 Hz GPS	8320 ± 1963	747 ± 387	33 ± 17	DNE	Physical demands of collegiate female soccer players during competition are similar, regardless of their position
Wells et al. [30]	9 NCAA DI F: 4 MF: 5 D: 5	RA across 17 regular season games & 4 postseason games	10 Hz GPS—Catapult Systems	Regular season: 7482 ± 959	Regular season: 86 ± 81	DNE	Sig ↓ high-speed and high-intensity runs during the second half of the postseason compared with the regular season**	Results indicate that additional minutes played during the postseason were primarily performed at lower intensity thresholds, suggesting running performance during postseason competitions may be compromised with greater playing time
Vescovi & Favero [29]	113 NCAA DI from 9 teams F: 33 MF: 45 D: 35	RA across 1 regular season	5 Hz GPS—SPI Pro, GPSports	First Half: F: 5232 ± 153 MF: 5186 ± 76 D: 4878 ± 74 * ----------------- Second Half: F: 5065 ±185 MF: 4939 ±121 D: 4618 ± 101 *	First Half: F: 146 ±26 * MF:87 ± 13 D: 131 ± 13 * --------------- Second Half: F: 193 ± 27 MF: 110 ± 18 D: 135 ± 15	DNE	D covered less total absolute distance than MF (first half), and MF and F (second half)	Moderate- and high-intensity distances cumulatively range from 2100 to 2600 m (26–28% total distance) in female college matches. It is suggested that substitution patterns have little impact on locomotor distribution
Ishida et al. [12]	17 NCAA DI F: 3 MF: 9 D: 5	RA across 1 competitive season	10 Hz GPS—OptimEye S5, Catapult Innovations	F: 6995.2 ± 3844.5 MF: 6869.8 ± 3973.0 D: 9339.4 ± 3860.6	F: 681.7 ± 492.0 MF: 594.9 ± 428.4 D: 734.2 ± 366.9	DNE	DNE	Internal and external training loads differ by position
McFadden et al. [10]	16 NCAA DI	RA across 1 competitive season	10 Hz GPS—Polar TeamPro System	8310.0 ± 1400.0	400.9 ± 157.6	13.8 ± 5.0	DNE	Tracking relative workload on an individual-basis may help to enhance the health and performance of athletes throughout the playing season
McCormack et al. [35]	10 NCAA DI	RA across 16 regular season matches	10 Hz GPS—Minimax 4.0, Catapult Systems	DNE	DNE	4.31 ± 3.51	Significant ↓ in high intensity running rate from Friday match to Sunday match	NCAA DI female soccer players run less total distance when matches are less than 48 h after a first, potentially requiring more intensive recovery strategies
Gentles et al. [2]	25 NCAA DII	RA across 17 regular season matches	GPS 5 Hz Zephyr^TM^ BioHarnesses	5480 ± 2350	DNE	DNE	DNE	This study details the demands of Division II women’s soccer match-play.
Jagim et al. [31]	25 NCAA DIII	RA across 22 regular season matches	GPS 10 Hz—Polar Team Pro	9807 ± 2588	1019 ± 552	15 ± 8	DNE	Significant and meaningful differences in movement kinematics were observed across position groups

* significant positional difference. ** significant difference across time. Pro = professional; F = Forwards; MF = Midfielders, D = Defender; CA = Central Attack; RA = Repeat Assessment; DNE = Did Not Evaluate; kph = kilometers per hour; HIR = High-intensity running; HR = heart rate; DI = NCAA Division I; DII = NCAA Division II; DIII = NCAA Division III.

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
