# Peer review of "Differences across Playing Levels for Match-Play Physical Demands in Women’s Professional and Collegiate Soccer: A Narrative Review"

_sports, 2022, doi:10.3390/sports10100141_

Round 1

Reviewer 1 Report

This manuscript aimed to provide a review of the available sport science literature in women's soccer from the professional ranks through the collegiate levels. Overall, this is a well-written narrative review and is well-organized. The majority of my comments below are rather minor in nature.

The introduction is a bit long for my taste in a narrative review. If the authors feel so inclined to shorten this section and get to the purpose of the paper sooner, that would be nice.

Ln 92: please cite the articles that you are referring to with "other relevant articles"

Results: There is no indication of how each metric is defined. I'm certain that the studies presented were not all uniform in defining thresholds for sprints, for examples, so providing a bit more detail in the results for those differences would be helpful.

Results: Several ranges for metrics are provided, and some by specific position. It would be helpful to view some of these data in graphical format rather than in words. It was difficult to make comparisons between the studies without a visual. The reader can get a little lost in the data.

Lns 248-251: substitution strategies would also affect sprint distance

Limitations: It is important to recognize two major limitations in the current literature: 1) very few, if any, of the studies present the data relative to time played in a match. Rather the data are all based on total volume for a game regardless of time played. 2) There are allusions to different metric definitions and thresholds within the literature, but further delineation regarding the use of absolute versus relative thresholds in the literature and the use of different percentages in defining zones is worth noting as a significant limitation in make comparisons.

Author Response

Thank you for your comments. I worked to shorten the introduction a bit (paragraph 5).

I cited the articles in Ln 92.

I added definitions for sprint in the results. Thank you for the suggestion to present results differently. I incorporated figures in the results.  

Yes, good point, thank you. I included substitution strategies in the list of what can affect sprint distance. (Ln 248-251)

Thank you for also summarizing the limitations to this study. I have included this information in the revised manuscript.  

Reviewer 2 Report

General comments

This manuscript aims at 1) consolidating available match-play data for each playing level to reveal similarities, discrepancies and gaps to shed light on the need to create external load profiles for female soccer athletes at specific playing levels and 2) suggesting future directions for research, which could help providing coaches with the data necessary to better design and implement evidence-based training programs. Aims are not that original. Yet, in spite of especially one relevant missing reference, specific and minor issues (detailed below), authors manage to fulfill sufficiently their aims.

Specific comments

Relevant reference missing:

https://pubmed.ncbi.nlm.nih.gov/35771861/

There should be some reference to professional soccer performance in abstract.

Minor comments

(line 155) Please, pay attention to correct in-text reff numbering;

(l221 and elsewhere throughout MS) please, check for correct in-text referencing;

(l235) need for bold?

(l333) there are no #44 or 45 reff listed.

Author Response

Thank you so much. I incorporated the Harkness-Armstrong et al., 2022 study. I added reference to professional soccer in the abstract and I went through and corrected some of the in-text citations and formatting.